# Cilia and Cancer: From Molecular Genetics to Therapeutic Strategies

**DOI:** 10.3390/genes14071428

**Published:** 2023-07-11

**Authors:** Pietro Carotenuto, Sergio A. Gradilone, Brunella Franco

**Affiliations:** 1Medical Genetics, Department of Translational Medical Science, University of Naples “Federico II”, 80131 Naples, Italy; 2TIGEM, Telethon Institute of Genetics and Medicine, 80078 Naples, Italy; 3The Hormel Institute, University of Minnesota, Austin, MN 55912, USA; sgradilo@umn.edu; 4Masonic Cancer Center, University of Minnesota, Minneapolis, MN 55455, USA; 5School of Advanced Studies, Genomic and Experimental medicine Program (Scuola Superiore Meridionale), 80138 Naples, Italy

**Keywords:** cilia, ciliome, cancer, ciliogenesis

## Abstract

Cilia are microtubule-based organelles that project from the cell surface with motility or sensory functions. Primary cilia work as antennae to sense and transduce extracellular signals. Cilia critically control proliferation by mediating cell-extrinsic signals and by regulating cell cycle entry. Recent studies have shown that primary cilia and their associated proteins also function in autophagy and genome stability, which are important players in oncogenesis. Abnormal functions of primary cilia may contribute to oncogenesis. Indeed, defective cilia can either promote or suppress cancers, depending on the cancer-initiating mutation, and the presence or absence of primary cilia is associated with specific cancer types. Together, these findings suggest that primary cilia play important, but distinct roles in different cancer types, opening up a completely new avenue of research to understand the biology and treatment of cancers. In this review, we discuss the roles of primary cilia in promoting or inhibiting oncogenesis based on the known or predicted functions of cilia and cilia-associated proteins in several key processes and related clinical implications.

## 1. Introduction

The surface of most eukaryotic cells is characterized by the presence of organelles with important motility and signaling hub functions: the cilia. These evolutionarily conserved extracellular structures project from the cell membrane to generate motility or function in cellular signaling [1].

Cilia consist of a dynamic structure, essentially composed of two functional units: (a) the basal body (BB), which forms the base of the cilium; (b) the axoneme, a protrusion that is anchored to the BB and extends out of the cell (Figure 1). The BB, a barrel-like microtubular structure located near the cell surface, forms the base of the cilium and arises from the mother centriole of the centrosome [2]. The centrosome as the major microtubule-organizing center of cells is involved in cell shape, polarity, and motility and has a crucial role in cell division [3,4]. The axoneme is the skeleton of the ciliary shaft. The cilium axoneme is characterized by a microtubule-based structure composed of nine microtubule doublets, anchored to the cell through a BB (Figure 1) [2,4].

Traditionally, vertebrate cilia have been classified on the basis of their axoneme structure into motile cilia (MCs) and primary cilia (PCs). In fact, the MC axoneme is characterized by a 9 + 2 microtubule pattern in which 9 peripheral doublets of microtubules surround 2 single centrally localized ones, while the PC has a 9  +  0 pattern with no central microtubules [5] (Figure 1). However, emerging findings have shown that this dichotomic classification is too restrictive. In fact, MCs with 9 + 0 microtubule pattern were found on the embryonic node deputed to generate fluid movement vital for left–right body axis specification [6], and specialized immotile 9 + 2 cilia were reported in the cells of the mammalian auditory and olfactory system [7]. A revised taxonomy of vertebrate ciliary subtypes has been proposed by Takeda et al. [4]. They added to the classical classification, based on morphology (structure of axoneme) and motility, a third category based on topography (number of cilia per cell) [4]. According to this classification, eight categories of cilia have been identified: classic PCs (solitary 9 + 0 non-motile cilia); classic nodal cilia (solitary 9 + 0 MC); multiple 9 + 0 non-MC; multiple 9 + 0 MC; solitary 9 + 2 non-MCs; solitary 9 + 2 MCs; multiple 9 + 2 non-MCs; conventional MCs (multiple 9 + 2 MCs) [4].

MCs are structurally hair-like protrusions, characterized for their locomotive function [8]. MC movement is possible because the extra pair of central microtubules is linked by radial spokes to the nine radial microtubules, which in turn are connected by dynein arms, which produce inter-microtubule movement [2,8]. MCs are present in single or multiple copies (>100) per cell and are found on the epithelial cells of the reproductive and respiratory tracts [8] and on the brain ventricles [9], where they beat in wave-like patterns to propel liquids and/or mucus [2,4,8,9,10]. Solitary MCs are present on the cells of the embryonic node to propel growth factors in a directional fashion for the establishment of left–right body axis [6]. Besides their locomotive function, MCs are also involved in the sensation of bitter taste to facilitate the beating of cilia as a defense system of the respiratory system [8,11]. Nodal cilia are specialized forms of MCs present at “embryonic nodes”, developmental structures with a critical role as embryonic organizers and determinants of left–right asymmetry [6].

The presence of a solitary, immotile PCs on the surface of almost all mammalian epithelial cells have been well described since the 1960s [12]. However, for decades, it was accepted that it had no function other than being a vestigial organelle.

Research over the past several decades has shed light on the important function of PCs as signaling hubs [12,13]. In fact, the vast majority of signaling pathways in vertebrates converge to PCs, which function as a “cells’ antenna”, mediating the transduction of external stimuli into the cell [14]. Moreover, PCs play key roles in multiple developmental pathways [12]. Specialized cilia are also present on the surface of sensorial cells and are responsible for photoreception [15], olfaction [16], hearing [17], and mechanosensation [18].

Recent studies have provided novel molecular insight for both PCs and MCs in regulating the communication between the cell cytosol and the external micro-environment [19,20,21]. High-resolution structural [22] and further functional studies [23,24,25,26,27] have demonstrated that cilia may mediate the communication between cells by functioning as molecular gates for extracellular vesicles (EVs) [19,28]. EVs are microparticles released from the cell into the extracellular space and delimited by a lipid bilayer [19,28]. The ciliary–EV axis has been shown to participate in intercellular communication through the transport of bioactive molecules (proteins, lipids, and non-coding RNA) or in the control of ciliary length and composition, notably through the discard of ciliary components [21,23,24,25,28].

With the advent of next-generation sequencing (NGS) technologies, the multi-omics approaches led to deciphering the molecular components of the ciliome [29,30,31]. The recent explosion of interest in cilia-related diseases has prompted systematic analyses of the ciliome (cilia-related genomic, proteomic, and transcriptomic data) and the regulation of cilia development and function [32,33]. The ciliome currently contains a comprehensive database of more than 2500 genes implicated in ciliary function [29,34,35]. By applying NGS approaches with a focus on the ciliome, new human disease genes have recently been identified [35].

Alterations in genes affecting the structure and function of cilia have been associated with an emerging class of genetic multisystemic human inherited conditions, known as ciliopathies. Ciliopathies range from largely organ-specific disorders, such as polycystic kidney disease (PKD) and nephronophthisis (NPHP), to pleiotropic disorders, such as cerebello-oculo-renal syndrome (CORS), Bardet–Biedl syndrome (BBS), Joubert syndrome (JBTS), Alström syndrome (ALMS), Jeune asphyxiating thoracic dystrophy (JATD), Meckel–Gruber syndrome (MKS), and oral-facial-digital type 1 syndrome (OFD1) [35,36,37].

Large-scale genomic studies provided accumulating evidence that PCs play an important role also in cancer formation and regulation [38]. Although MC dysfunction has been associated with some diseases, there is still no evidence that there is a connection with tumors originating from these organs [39].

Therapeutic approaches targeting PCs in cancer have been taken into consideration by several authors, as well as the use of PC components as biomarkers [40,41].

Here, we review the basics of cilia biology and discuss the meaning and significance of alterations of the ciliome in cancer and the consequent clinical implications. Given the increasingly emerging data regarding the influence of the PC in cancer progression, we will mainly focus on the implications in molecular oncology of the PC, rather than the MC, highlighting the role of PCs in cancer and suggesting directions for future research.

## 2. Role of Primary Cilia in Oncogenic Programs

The elucidation of cilia biology has highlighted the key regulatory functions of PCs in several events regulating cell growth and intercellular and intracellular communication. In particular, the main role of PCs in regulating the cell cycle and molecular signaling is of considerable importance in oncology, since their deregulation constitutes the molecular basis of several cancers.

### 2.1. The Functional Link between Cilia and Cell-Cycle-Related Oncogenic Programs

Uncontrolled cell proliferation and deregulation of the cell cycle are hallmarks of cancer cells and neoplastic development. In this section, we describe how the PC works as a key player in the regulation of the cell cycle and of related oncogenic programs.

The functional interplay between the PC and the cell cycle was first recognized with the observation that PCs are disassembled before mitosis [42]. Indeed, in most mammalian cells, the process of PC assembly, named ciliogenesis, occurs in the post-mitotic G0/G1 phase of the cell cycle, while PC disassembly takes place before mitosis, in strict association with the centriole cycle [42]. Ciliogenesis and the cell cycle are closely related molecular processes that share a common macro-molecular complex, the centrosome. The centrosome is a microtubule-based structure present at the base of PCs during ciliogenesis, acting as a template for axonemal growth, while during mitosis, it serves as a microtubule-organizing center (MTOC) for the generation of the mitotic spindle [42,43]. The centrosome is typically composed of two cylindrical microtubule-based structures termed centrioles, which recruit a matrix of associated pericentriolar material. The centriole has a dual life, existing not only as the core of the centrosome, but also as the BB [44,45].

It is commonly accepted that ciliogenesis and cell division are mutually exclusive processes since each depends on the exclusive use of the centrosome [8]. In dividing cells, the PC is disassembled before cell division and centrioles are inherited by daughter cells, in which they act as templates for the next generation of cilia [46,47].

As ciliogenesis requires a complex program of macromolecular synthesis and assembly, it must be carefully regulated. Given the double function of the centrosome, defects in cilium formation may affect the cell cycle, or inversely, alterations of the cell cycle affect ciliogenesis. Supporting this idea, many highly proliferating cancer cells lack PCs, suggesting that the absence of cilia may drive mitosis and, consequently, cell proliferation [40].

Several molecular regulators of centrosome function in mitotic spindle assembly also play a critical role in ciliogenesis [46,47]. Centrosome maturation is a crucial step for bipolar mitotic spindle assembly, and this process is characterized by the recruitment and re-organization of additional pericentriolar material, the phosphorylation of centrosomal proteins, and a dramatic increase in microtubule nucleation and the anchoring capacity of the centrosome (Figure 1).

A number of protein kinases that act as key molecular players in mitotic events are crucial for the maturation of centrosomes, including PLK1, AURKA, NEK2, and KIF2/24 (Figure 1). These proteins act by phosphorylating centrosomal proteins and are implicated in several oncogenetic programs.

PLK1 plays a pivotal role during the M phase of the cell cycle [48,49]. Several mitotic events are regulated by this kinase, which phosphorylates substrate proteins on centrosomes, kinetochores, the mitotic spindle, and the midbody, including cyclin-dependent kinases (Cdks), mitotic kinesin-like motor protein (MKLp), and centrosome components KIZ, NEDD1, and NINL [50]. PLK1 kinase activity is required to promote PC disassembly before mitotic entry [51]. It is highly overexpressed in various human cancers and is thought to promote tumorigenesis [52].

Aurora kinase A (AURKA) is a centrosomal mitotic kinase that regulates S-phase entry. Prior to mitosis, it localizes to the BB and is activated by the scaffold protein human enhancer of filamentation 1 (HEF1) and calmodulin (CaM) in the presence of calcium [14]. The HEF1-Ca^2+^/CaM–AURKA complex, in turn, activates deacetylase histone deacetylase 6 (HDAC6), which destabilizes axonemal microtubules, inducing PC disassembly [53]. AURKA was found to be upregulated in non-ciliated ovarian and renal cell carcinoma cells [53,54,55,56], and HDAC6 inhibition restored PCs in chondrosarcoma and cholangiocarcinoma cancer cells, suppressing cell proliferation and invasion capacity [57]. Similarly, HEF1 overexpression has been associated with cell migration and cancer progression in several tumors, including breast cancer and melanoma [58].

Moreover, PLK1 and AURKA participate in common molecular processes regulating PC disassembly. Indeed, the activation of the non-canonical Wnt pathway induces the formation of the PLK1-disheveled segment polarity protein 2 complex (PLK1-DVL2), which activates AURKA through the stabilization of HEF1, thus inducing cilium disassembly [48].

NEK2 is another important regulator of both centrosomes and BBs [59]. NEK2 exerts its role in the disassembly of the axonemal microtubules by phosphorylating kinesin family member 24 (KIF24), a member of the kinesin superfamily of microtubule-based motor proteins, which stimulates its microtubule-depolymerizing activity and prevents the formation of cilia in proliferating cells [59]. NEK2 and KIF24 have been found to be overexpressed in several cancers. It has been shown that the inhibition of these proteins in breast cancer cell lines lacking cilia restores ciliation, thereby reducing cancer growth [59,60].

Additionally, cilia are devoid of protein synthesis and rely on efficient intraflagellar transport (IFT) for the supply of cilia components and for ciliogenesis. IFT is thought to be the predominant pathway to move proteins into and within cilia [61,62]. IFT multisubunit complexes mediated the axonemal outgrowth, turnover, and disassembly, thus IFT proteins are the limiting factors, together with tubulin availability, for cilium formation. The core elements of the IFT machinery are two sub-complexes, A and B, linked to a BBSome complex bearing cargo [61,62]. As such, IFT can influence cell cycle progression and plays an important role in vertebrate development, signaling, cellular motility, sensory transduction, and homeostasis. Accumulating evidence shows that defects in some IFT proteins, which result in impaired ciliogenesis, may determine the progression of several cancers [63].

### 2.2. The Functional Link between Cilia and Cancer-Related Signaling Networks

Proliferative signaling pathways involved in normal cellular growth and tissue development are frequently deregulated during tumor initiation, progression, and therapeutic response. The PC is a key mediator of altered proliferative signaling due to its function in sensing signals from the extracellular environment. In recent years, increasing evidence has attributed the PC with the role of a molecular hub, transducing a plethora of signals from several well-characterized signaling networks including the Hedgehog, Wnt, and PDGF pathways [12]. In this section, we will describe the main signaling pathways converging on the PC, attempting to comprehensively discuss their relationship with cancer (Figure 2).

#### 2.2.1. Hedgehog Pathway

The *Hedgehog* (HH) signaling system promotes tumor growth by serving as an oncogenic driver. HH signaling is mediated by various proteins secreted by epithelial or tumor cells and exert their function by binding the receptor Patched (PTCH1), which is localized on the PC membrane of either the HH-secreting or neighboring cells [64]. In the absence of HH binding, PTCH1 transduces repressive signals that sequester a second protein, Smoothened (SMO), in intracellular vesicles. HH binding to PTCH1 causes PTCH1 to be trafficked outside the PC, allowing SMO to translocate into the cilia, where it activates a transcriptional program dependent on zinc finger protein GLI effectors [64]. Additional cellular proteins act as modulators of HH: for example, the G-protein-coupled receptor (GPR161) has recently been defined as an HH regulator with cancer relevance [65,66,67]. While GPR161 has been reported to drive oncogenic programs in breast cancer [66], it acts as a negative regulator of the HH pathway in medulloblastoma (MB), where GPR161 negatively influences MB progenitor proliferation [66,67]. Activated GLI proteins move to the nucleus, where they bind and activate the transcription of a suite of genes that control processes relevant to tumor growth and resistance to treatment [68,69]. Among these genes, there are cyclin D1 (*CCND1*) and *MYC* (controlling cancer cell proliferation), zinc-finger protein *SNAI1* (controlling epithelial–mesenchymal transition (EMT)), *BCL2* (controlling survival), transcription factor *SOX2* and homeobox protein *NANOG* (controlling stem cell identity), and angiopoietin (*ANGPT*) 1 and 2 (controlling angiogenesis) [68,69].

Another mechanism by which GLI proteins are regulated in a cilia-dependent manner is through the protein Suppressor of Fused (SUFU). SUFU is known to be a negative regulator of HH signaling that localizes to cilia tips and also has broad cancer relevance including a role in Gorlin syndrome [70]. Moreover, in human cancers, genetic alterations of HH-related genes such as *PTCH1*, *SMO*, and *SUFU* significantly influence cilia-dependent tumorigenesis [68,70].

#### 2.2.2. Notch Pathway

Notch has been recognized as an ancient and highly conserved signaling pathway regulating cell fate determination and tissue homeostasis [71]. In cancer, the oncogenic activation of Notch proteins has been observed in lung, breast, and several other tumor types [72]. Notch signaling drives a plethora of cancer-related programs, including stem cell proliferation, invasion, metastasis, and angiogenesis [72]. The activation of Notch requires the cleavage of ligand-bound activated Notch by the γ-secretase complex, which is localized proximal to the BB, leading to release of the Notch intracellular domain (NICD). The NICD translocates to the nucleus, where it forms a complex with the transcription factor CSL and induces MYC, CCND3, the HES family bHLH transcription factor 1 (HES1), and other genes. The molecular link between Notch signaling and cilia has been demonstrated by Ezratty et al., who reported that the knockdown of IFT proteins causes defects in Notch signaling and impairs progenitor cell differentiation during skin development [73]. Furthermore, the interplay with other signaling networks is fundamental during epidermal and neural differentiation, where Notch signaling sensitizes progenitor cells to respond to HH, causing an increase of cilia length [73].

#### 2.2.3. Wnt Pathway

Wnt signaling is implicated in a large number of developmental and disease processes. Several studies have contributed to elucidating the role of canonical and non-canonical Wnt signaling in cancer [74]. Aberrant Wnt signaling has been implicated in several steps of carcinogenesis, including cancer stemness, metastasis, and immune surveillance [74]. The hypothesis that the PC was necessary for Wnt signal transduction was supported by reports showing that components of the Wnt pathway, including Inversin/Nephrocystin2, Vangl-2, Gsk3-β, and Apc, are located in proximity to the PC or BB [75]. Altered Wnt signaling contributes to the loss of cell polarity, which is associated with unbalanced proliferation in numerous epithelial cancers. Canonical Wnt signaling depends on the activation of a β-catenin effector. Non-canonical Wnt signaling involves regulation of planar cell polarity (PCP) effectors and proteins such as phospholipase C (PLC), which regulate calcium responses [76]. Several studies have focused on the connections between cilia and the Wnt pathway, demonstrating that the ciliome may have an important role in regulating the activation and regulation of both canonical and non-canonical pathways.

Nephrocystin2 localizes to cilia, physically interacts with the core component of the Wnt pathway, dishevelled (Dvl), and inhibits the ability of Dvl to drive the activation of Wnt canonical signaling [77]. Several genes associated with Bardet–Biedl syndrome (BBS1, BB4, and MKKS), which encode BB proteins, interact with components of Wnt signaling. Their suppression results in stabilization of β-catenin with concomitant hyperactivation of Wnt response in cultured cells. In cell culture, knockdown of the kinesin motor KIF3a, an essential player in ciliogenesis, also results in altered response to exogenously supplied Wnt3a [78]. Moreover, disruption of PCs in mice harboring mutations in *KIF3a*, *IFT88*, or oral-facial-digital syndrome type 1 (*OFD1*) has been shown to result in a marked increase in cellular responses to canonical Wnt pathway activation [78].

#### 2.2.4. Receptor Tyrosine Kinases and Other Membrane-Associated Kinases

The dysregulated activity of receptor tyrosine kinases (RTKs) is one of the most-common oncogenic driver mechanisms. Various studies have shown that cancer-relevant RTKs including insulin-like growth factor 1 receptor (IGF1R), epidermal growth factor receptor (EGFR), TEKs (angiopoietin receptors), fibroblast growth factor receptor (FGFR), platelet-derived growth factor receptor α (PDGFRα), and TGF-β receptor (TGF-βR) are located within or proximal to PCs [79].

Schneider et al. first demonstrated the localization of RTKs to PCs [80]. They showed that PDGFRα localizes to the PC in fibroblasts and that PDGFRα activation is strictly dependent on the presence of PCs. Notably, aberrant PDGFRα signaling has been associated with several pathologies including gastrointestinal stromal tumors [81].

EGFRs extensively regulate cellular processes during development and in tissue homeostasis [82]. Amplification of the EGFR gene and mutations of the EGFR tyrosine kinase domain have been recently demonstrated to occur in carcinoma patients [82,83]. EGFR signaling in PCs was first reported in studies on mechanosensation in cultures of kidney epithelial cells. EGFR has been associated with the PC also in astrocytes and neuroblasts [84]. In airway smooth muscle cells [85], EGFR was suggested to play a major role in mechanosensation and cell migration through the interaction with integrins and the cilioproteins polycystin 1 and 2 (PKD1 and PKD2) in the PC [85]. Recently, it has been reported that EGFR suppresses ciliogenesis by directly phosphorylating the deubiquitinase USP8 in RPE1 cells [86].

FGFR receptors have a well-recognized function in the regulation of PCs. Aberrant FGFR activity produces abnormal cilia with deregulated signaling, which contributes to the pathogenesis of FGFR-mediated genetic disorders [87]. FGFR alterations are also found in cancer [88], raising the possibility of cilia involvement in neoplastic transformation and tumor progression. Inactivation of the FGF-receptor FGFR1 or its FGF ligands leads to shorter cilia in zebrafish and Xenopus [89]. In mammals, FGF signaling regulates the length of primary cilia in skin, lung, kidney, and liver cells, human embryonic and induced pluripotent stem cells, embryonal fibroblasts, and mesenchymal cells [89]. In addition, human skeletal dysplasias such as achondroplasia caused by activating FGFR3 mutations are characterized by abnormal cilia [87,89].

Finally, TGF-βRs have been shown to localize to PCs, and TGF-βR-related effector signaling components including ERK1/2 and SMAD3 accumulate at the ciliary based on TGF-β-induced activation [90,91]. Notably, signaling by the TGF-βRs and the PC-localized HH pathway promotes cell migration and tumor metastasis [90].

#### 2.2.5. Hippo Pathway

Hippo is a conserved signaling pathway regulating organ size in *Drosophila melanogaster* and mammals. Dysregulation of this pathway is related to cancer development [92]. The downstream transcriptional activators such as YAP/TAZ are the major effectors of Hippo signaling, which play a variety of roles in regulating cell–cell contact inhibition, epithelial–mesenchymal transition, development, proliferation, and differentiation. Hippo signaling is commonly considered an oncogenic pathway [92]. According to several reports, YAP and TAZ can play different roles in ciliogenesis [92]. Recently, NPHP4, a ciliary protein mutated in the form of nephronophthisis, has been reported to act as a potent negative regulator of the tumor-suppressive Hippo pathway [93]. Interestingly, several NPHP-related proteins have been found to be upregulated in various tumors, including breast cancer [94], pancreas carcinoma [95], and colorectal cancer [96].

#### 2.2.6. DNA Damage/Repair Pathway

The deregulation of DNA damage/repair pathway is associated with the initiation and progression of cancer [97]. Work from several groups has highlighted the functional link between the DNA damage response (DDR) pathway and PCs [98]. Indeed, the role of PCs in the maintenance of the DDR is supported by findings that centrosomes contain several components belonging to the DDR [98,99]. These factors include the DNA repair proteins BRCA1, BRCA2, PARP1, and NBS1 and other molecular players that initiate repair responses such as ATM, ATR, cell cycle checkpoint, and TP53, which also colocalize with the BB. The existence of an interplay between DDR-related players and centrosome physiology is supported by several reports showing that TP53 and PARP1 deficiency and BRCA1 mutations determine an altered centrosome biogenesis, which leads to an aberrant number of PCs [98]. Additionally, centrosomal protein 164 (CEP164) has been reported to interact with the DDR-related protein ATM and to be phosphorylated in response to DNA damage, and its downregulation is concomitant with decreased phosphorylation of other DDR proteins such as CHK2 and CHK1 [98]. Oral–facial–digital syndrome Type I (OFD1), a gene with a well-characterized role in the centrosome/basal body/cilia network, can negatively impact important nuclear events: chromatin plasticity and DNA repair [100]. In fact, OFD1 patient-derived cells showed pronounced defects in the double-strand break-induced histone modification, chromatin remodelling, and double-strand break (DSB) repair [100]. Finally, DNA damaging factors influence centrosome function by causing distortion and duplication of pericentriolar material [98,99].

#### 2.2.7. Autophagy Network

Autophagy is an evolutionarily conserved process used by cells to sequester intracellular components, to degrade their content, and recycle nutrients back to the cytoplasm. Autophagy is a tightly regulated pathway that maintains cellular homeostasis. Dysfunction of autophagy has been linked to a variety of human diseases including cancer and degenerative and immune disorders. The mutual crosstalk between ciliogenesis and autophagy has been established by several reports showing that both ciliogenesis and autophagy are commonly stimulated by serum starvation in cultured cells (reviewed in [101,102]). Moreover, autophagy is an important key step in the biogenesis of PCs by controlling the degradation of ciliary proteins. The term “ciliophagy” has been proposed for the degradation of ciliary proteins by autophagy [103]. The first evidence of the role of autophagy in regulating ciliogenesis was provided by studies on the *OFD1* gene [101,102,103]. The OFD1 protein localizes to centrosomes/basal bodies and is necessary for the formation of distal appendages of the mother centriole, a process required for PC formation [104,105]. Tang et al. and, more recently, Morleo et al. demonstrated that autophagy-mediated OFD1 degradation at the centriole satellites is needed for ciliogenesis [102,106]. Depletion of OFD1 by RNA interference dramatically increased cilia formation in murine embryonic cells and restored ciliogenesis in MCF7 breast cancer cells that originally lacked cilia [106]. Interestingly, excessive accumulation of OFD1 due to dysfunctional autophagy has been associated with endometrioid and oropharyngeal carcinoma [107,108].

Moreover, the intraflagellar transport (IFT) complexes have been implicated in a reciprocal interplay between autophagy and ciliogenesis [109]. In particular, intraflagellar transport protein 20 (IFT20) is required for the biogenesis and function of lyso-autophagy by controlling the lysosomal targeting of acid hydrolases [110]. A repressor role for IFT20 in breast cancer cell migration has been recently described [111].

Furthermore, HDAC6 has been identified as an important cytoplasmic tubulin deacetylase that influences mitosis and chemotaxis through the regulation of tubulin stability and as an enhancer of autophagosome–lysosome fusion [110]. Overexpression of HDAC6 correlates with tumorigenesis, and inhibitors of HDAC6 were proposed as anti-cancer therapeutic approaches [40,112].

MTOR is a serine/threonine kinase that participates in different cellular processes such as cell size regulation, metabolism, growth, proliferation, and survival. PCs regulate mTOR through the activation of the ciliary protein LKB 116. It has been reported that the activation of LKB1 leads to the inhibition of migration and invasion in normal ciliated cholangiocytes [113]. Moreover, LKB1 depletion results in pancreatic cystic tumor formation in mice, suggesting that the interplay between the LKB1-mTORC1 pathway and PCs may be an important regulator of tumorigenesis in pancreatic cancer [114].

While autophagy can control ciliogenesis, defects in PCs also have a profound impact on autophagy activity. The link between PCs and autophagy is conceivable since autophagy is responsive to mTORC1 activity, which is controlled by PCs. In normal ciliated cells, flow stress increases autophagy activity and reduces cell size [102].

Experimental evidence suggests that the interplay between autophagy and PC may have a key role in cancer development since the concomitant loss of PCs and autophagy upregulation are frequently observed in cancer [40,115]. However, further studies are needed to decipher the functional role of the autophagy/cilia axis in cancer.

#### 2.2.8. The Polycystin Signaling

Polycystin-1 and polycystin-2 are large transmembrane proteins encoded by the *PKD1* and *PKD2* genes. They localize at cilia and affect multiple downstream signaling pathways [116,117,118]. Both proteins also function as mechano-sensors and mediate the PC-sensing abilities of fluid secretion. Mutations in *PKD1* and *PKD2* cause autosomal dominant polycystic kidney disease (ADPKD), a progressive inherited disorder in which the renal tissue is gradually replaced with fluid-filled cysts, giving rise to chronic kidney disease (CKD) and progressive loss of renal function [116,117].

Emerging evidence has demonstrated the involvement of *PKD1* and *PKD2* in several cancer hallmarks, including proliferation, apoptosis, and interaction with the tumor micro-environment, thus suggesting the potential contribution of this family of proteins to tumorigenic processes. A study conducted by Gargalionis A. and colleagues investigated the functional role of *PKD1* and *PKD2* as the main players in mechano-transduction in colorectal cancer (CRC) [119]. They provided in vitro and in vivo data demonstrating the link between polycystins and CRC progression [119]. Clinical relevance in CRC was also assessed, and *PKD1* and *PKD2* overexpression was associated with poor survival in a cohort of 190 CRC patients [119]. On the other hand, other groups proposed a potential tumor-suppressing role for *PKD1*, demonstrating that the overexpression of *PKD1* in lung cancer cells, CRC, and hepatocellular carcinoma inhibited the invasion and migration of tumor cells [120]. In melanoma B16 mice cells, the silencing of *PKD2* by siRNAs (small interfering RNA) resulted in significant suppression of intercellular adhesion [121].

Moreover, the activation of signaling pathways involved in sustaining cancer proliferation, including EGFR, HER2, B-RAF, ERK, mTOR, AKT, and SRC, have been identified in association with ADPKD, accumulating evidence that polycystins may be involved in cancer development and progression [118]. Since ADPKD and cancer present several common molecular features, several authors proposed that mutations in *PKD1* or *PKD2* might be relevant in predisposing patients to kidney cancer [122,123]. Studies conducted by the group of Hajj, P. et al. and Orskov, B. et al. first investigated the prevalence of renal cell carcinoma (RCC) in patients with ADPKD, describing increasing incidence of RCC in two different cohorts, but these studies had several limitations due to sampling and the number of RCCs [124,125]. A more-recent study analyzed the incidence of cancer in 10,166 kidney transplant recipients comparing patients with and without ADPKD [126]. Interestingly, this study reported a significantly lower risk of kidney cancer in ADPKD patients [126].

## 3. Primary Cilia Defects in Cancer: Implication for Molecular Oncology

Several authors proposed that the PC may work as a tumor suppressor organelle [40,57,127,128]. This hypothesis is supported by two key facts: (1) the PC transduces cancer-related signaling pathways [12,13]; and (2) PCs are absent in different types of tumors [40,41,57,113,127]. In the following section, the link between PCs and cancers will be discussed and the clinical relevance of PCs in several kinds of tumors further developed.

### 3.1. Brain Cancers

In medulloblastoma (MB), a dual function for PCs has been described by Han and colleagues [68]. Genetic ablation of PCs blocked medulloblastoma growth when this tumor was driven by a constitutively active SMO, an upstream activator of HH signaling. In contrast, removal of PCs was required for medulloblastoma growth by a constitutively active GLI2, a downstream transcription factor. Remarkably, the analysis of PC expression in 111 MBs revealed that the presence or absence of PCs was associated with specific variants of medulloblastomas. PCs were found in MBs with activation in HH or Wnt signaling, but not in most MBs in other distinct molecular subgroups [129]. Recent advances suggested that driver mutations in several genes belonging to the HH pathway, including ciliary-related genes, may be involved in MB etiopathogenesis. Interestingly, germline mutations in *SUFU*, *PTCH1*, and *GPR161* have been observed in heritable predisposition to MB, such as Gorlin syndrome [68,70]. Based on these observations, it has been suggested that the oncogenic or tumor suppressor function of PCs in MB may depend on the initiating oncogenic event [129].

Aberrant ciliogenesis is commonly found in cells derived from astrocytomas/glioblastomas (GMBs), and this deficiency likely contributes to the phenotype of these malignant cells [130].

A recent study analyzing the transcriptome of glioma patients revealed that those displaying a high expression of PC-associated genes had significantly poorer prognosis compared to the rest of the cohort, independent of the grade or other prognostic biomarkers [131].

In GBM, PCs are highly associated with tumor progression and therapeutic resistance. Cilia-related signaling pathways, including SHH, cell cycle-related kinase (CCRK), and HDAC6, are closely connected with the proliferation, malignant development, and therapeutic resistance of GBM. Other cilium-related pathways, including the lysophosphatidic acid receptor 1 (LPAR1) and pericentriolar material 1 (PCM1) pathways, inhibit the proliferation and development of GBM cell lines. The EGFR, PDGFRα, MGMT, and isocitrate dehydrogenase 1 (IDH1) pathways promote GBM therapeutic resistance, which is associated with or modulated by the assembly and disassembly of PCs [132].

### 3.2. Skin Cancers

Basal cell carcinoma (BCC), one of the most-common skin cancers, is characterized by dysregulation of the HH pathway. Mutations that lead to the upregulation of HH signaling are frequently associated with the development of BCC. For instance, mutations in the tumor suppressor gene PTCH are implicated in the growth of sporadic BCCs and those that develop due to Gorlin syndrome [133]. As previously described, HH signaling and PC functions are strictly correlated. In a study by Kuonen et al., an increased number of mutations in PC-related genes and the loss of PCs in BCC were described. The authors also showed that the loss of PCs correlated with lower HH and higher Ras/MAPK pathway activation [134].

A study conducted by Kim et al., involving the analysis of 62 cases composed of typical melanocytic nevi, in situ melanoma, invasive melanoma, and metastatic melanoma, described a significant loss of PCs in advanced stages [135]. Moreover, in melanoma, the oncogene enhancer of zeste homolog 2 (*EZH2*) has been shown to be a main molecular player in the silencing of ciliary genes during melanoma development [136]. The same authors also demonstrated that the loss of PCs enhances the pro-tumorigenic effects of Wnt/β-catenin signaling [136]. Moreover, in melanoma, *BRAF* mutations are linked to a high expression of *EZH2*, which is associated with melanoma progression, worse patient survival, and resistance to MAPK inhibitors [137]. It has been demonstrated also that therapeutic strategies targeting *EZH2* or its downstream targets, such as the ciliary-related oncogene *PLK1*, in combination with BRAF inhibitors are potential novel therapeutic options in melanomas with *BRAF* mutations [138].

### 3.3. Gastrointestinal Cancers

Notably, an analysis of ulcerative colitis and colorectal cancer (CRC) patient biopsies showed a lower number of PCs on colonic fibroblasts in pathological versus surrounding normal tissue [139,140]. Sénicourt and colleagues analyzed the presence of PCs in CRC cells and tissues and identified PC-like structures in 58% of cancers at all stages, but not in adenomas, suggesting that PC appearance may occur relatively early in the carcinogenesis cascade, but it is not a feature associated with benign intestinal lesions [141].

Using CRC cell lines, they also found a significant correlation between the presence of PCs and the expression of the final HH effector, GLI1, and provided evidence of a functional link between the PC and GLI1, by demonstrating the recruitment of the SMO receptor to the membrane of PCs [141]. Using murine models, Rocha et al. demonstrated that tubulin glycylases are required for PC formation, the control of cell proliferation, and tumor development in CRC [142].

In cholangiocarcinoma (CCA), there is a reduction of PCs in vivo and in vitro in human patient samples and cells, respectively [41]. The deciliation of normal cholangiocyte cells using drugs such as chloral hydrate or using gene silencing of ciliary proteins such as IFT88 induces proliferation, anchorage-independent growth, and invasion [41]. Furthermore, it also induces the activation of the HH and MAPK pathways, which are normally negatively regulated by cilia and which are both involved in establishing malignant cholangiocarcinoma phenotypes [41]. The mechanisms leading to deciliation in these tumor cells, as well as the consequences of such a loss remain understudied.

Genes belonging to the *HDAC* family have been shown to work as key molecular players in mechanisms regulating the deciliation in CCA [57,128]. HDACs can inhibit the acetylation of target proteins and eventually lead to protein degradation. Gradilone et al. reported that α-tubulin, a ciliary axoneme protein, is targeted by HDAC6, which inhibits cilia formation [57,113].

It has been demonstrated that dysregulation of microRNAs (miRNAs) in CCA can promote several oncogenic programs [143,144,145,146,147], and a microRNA-dependent regulation of PC functions has been reported [148]. In fact, microRNAs are responsible for overexpression of HDAC6 in CCA leading to ciliary disassembly. Moreover, targeting HDAC6 in CCA cells decreases the tumorigenic phenotype in a ciliary-re-expression-dependent manner in vitro and in an animal model of CCA [57]. Similarly, other epigenetic regulators are suggested to modulate both ciliogenesis and PC functions. For example, a key role for SIRT1 as a tumor promoter that inhibits PC formation in CCA cells and induces cell proliferation has been demonstrated by the same group, who also provided a rationale for clinically exploring the use of SIRT1 inhibitors in CCA [128].

Studies performed on pancreatic adenocarcinoma (PDAC), using human tissue specimens, the PANC-1 cell line, and murine models, demonstrated the absence of PC [149,150]. Two studies assessed the presence and distribution of PCs in PDAC patients [149,150]. In particular, Seeley et al. studied PC expression in 17 patients with PDAC, reporting that the loss of PCs occurs in the earliest stages of PDAC development and that PC absence is independent of the proliferation status [149]. Moreover, they also showed that overactivation of the KRAS pathway arrests ciliogenesis [149]. Notably, the EGFR/KRAS axis is commonly recognized as an oncogenic driver of both ciliogenesis and tumorigenesis [83,151]. In the study of Emoto et al., PCs were identified in 100 PDAC patients who received no therapy prior to initial surgery [152]. A statistically significant association between overall survival (OS) and tumor size, grade, lymph node metastasis, and the presence of PCs was found [149]. In their report, they further showed that PC expression constitutes an independent poor prognostic factor of OS in pancreatic cancer [152].

It has been demonstrated recently that the PC is gradually lost in the epithelium during pancreatic carcinogenesis, and this loss is accompanied by a gain of PCs in the surrounding stroma [153]. Notably, the heterogeneity of the tumor microenvironment (TME) is a main determinant of PDAC progression and therapy resistance [154]. Kobayashi and colleagues identified HDAC2 and KRAS as the main regulators of ciliogenesis in PDAC [155], showing that the inhibition of both decreases the expression of AURKA, thus promoting PC restoration [155].

Deng et al., instead, investigated the biological functions of PCs during malignant transformation. This study demonstrated that the mevalonate (MVA) pathway is activated upon inactivation of ciliogenesis and may accelerate oncogene-induced transformation of normal cells both in vitro and in vivo [156].

### 3.4. Genito-Urinary and Endocrine Cancers

Ovarian cancer presents a significantly reduced numbers of PCs. The reduction of cilia in these cells was not due to a failure in growth arrest and correlated with persistent centrosomal localization of AURKA and with decreased HH signaling and PDGRFα expression [80].

In breast cancer, the reduction of ciliated cells has been described during cancer progression [157,158], and it has been directly linked to the downregulation of ciliary genes regulating the PC structure [157,158].

Furthermore, split ends (SPEN), an estrogen receptor co-repressor, is co-expressed with a dataset of ciliary genes regulating ciliary biogenesis. SPEN positively regulates PC formation and cell migration in breast cancer, possibly via the transcriptional regulation of the ciliogenic transcription factor *RFX3* [159]. In addition, in breast cancer patients, high expression of SPEN correlates with early metastasis and the presence of PC exclusively in patients with hormone-receptor (HR)-negative disease [159]. Furthermore, PCs were identified only in breast cancer basal B subtype epithelial cells, whereas they were absent in luminal and basal A subtype cells [157,158].

Moreover, PC frequency and length are decreased in all stages of prostate cancer, from early preinvasive lesions to invasive stages. In this tumor type, the absence of PCs correlates with increased levels of Wnt signaling [160].

Renal cell carcinoma (RCC) shows the loss of PCs connected with a downregulation of the von Hippel–Lindau (*VHL*) tumor suppressor gene. VHL participates in the control of oxygen levels and microtubule stabilization through the activation of HIFα and AURKA, and consequently activation of HDAC6. Mutations of the tumor suppressor gene *FLCN*, a protein with a ciliary localization, are linked with Birt–Hogg–Dubé syndrome, which manifests with renal cysts and predisposes to an increased risk of kidney tumor development. Tuberous sclerosis is caused by mutations of the tumor suppressor genes *TSC1* or *TSC2* and can lead to renal manifestations such as renal cell carcinoma and renal cystic disease. TSC1 is located at the base of PCs, and its downregulation produces increased ciliary length. In a study conducted in 110 patients of various RCC subtypes, a severe reduction of cilia frequency in various RCC subtypes was observed, suggesting that PC loss is a common event in renal tumorigenesis and implying that cilia loss is part of a sequence of events leading to renal tumor development [161].

Thyroid cancer (TC) is the most-common endocrine cancer and has a rapidly increasing incidence, but relatively stable mortality. The main histological subtypes of TC are papillary thyroid cancer (PTC), follicular thyroid cancer (FTC), poorly differentiated thyroid cancer (PDTC), anaplastic thyroid cancer (ATC), and medullary thyroid cancer (MTC) [162]. The first four types originate from thyroid follicular epithelial cells, while MTC arises from thyroid parafollicular cells. PCs are well preserved in PTC and FTC, and their frequency and length appear similar to those of normal thyroid follicles. Interestingly, defects in PC formation have been observed in ATC. Additionally, oncogenic alterations, coupled to specific intracellular downstream signaling pathways, lead to the development of different subtypes of TC. PCs, as a mediator of these signaling pathways, regulate TC development. Alterations in PC number influence the communication between TC cells and TME, which in turn affects the therapeutic response and prognosis of TC. The loss of primary cilia results in apoptogenic stimuli, which are responsible for mitochondrial-dependent apoptotic cell death in differentiated thyroid cancers [163]. Interestingly, oncogenic alterations including fibroblast growth factor receptor 2 (*FGFR2*) gene fusions with the ciliary gene *OFD1* were reported in TC [164,165]. FGFR2-OFD1 induced transformation in vitro, which was abolished by FGFR kinase inhibitors [164,165]. Furthermore, mice lacking PCs on the thyroid due to a thyroid-follicular-epithelial-cell-specific deletion of the ciliary gene intraflagellar transport protein 88 (*IFT88*) showed follicular cells with the malignant phenotype and developed papillary solid proliferative thyroid follicles with malignant features [163].

### 3.5. Sarcomas

Ciliary dysfunction in rhabdomyosarcomas (RMSs) and chondrosarcomas have been reported. De Andrea et al. revealed deficient ciliogenesis in the early phases of chondrosarcoma genesis [166]. Furthermore, the abnormal expression of HDAC6 and IFT88 has been described in chondrosarcoma tissues. The inhibition of HDAC6 causes PC restoration and suppresses the proliferation and invasion of chondrosarcoma cells [167]. Loss of PCs potentiates BRAF/MAPK pathway activation in rhabdoid cancers [168]. It has been suggested that primary ciliogenesis [168] and MAPK pathway activation [169] may contribute to rhabdoid cancer progression and, therefore, may constitute a novel therapeutic target.

Concerning rhabdomyosarcoma, it has been shown that PCs may contribute to the hyper-activation of HH signaling in a subset of RMSs [170]. Recently, an important role in controlling RMS cell growth was attributed to the ciliary protein ADP ribosylation factor like GTPase 6 (Arl6), which acts by regulating both cilia assembly and HH signaling in RMS [171].

## 4. Molecular Oncology of Primary Cilia: Clinical Implications

PCs serve as cell surface biomarkers associated with a growing number of pathologies, including cancer (Table 1).

### 4.1. Brain Tumors

In brain tumors such as glioblastoma or medulloblastoma, the study of the ciliome has provided a detailed view of the genomic alterations and signaling pathways affected. It has been proposed that this information may be used as diagnostic, prognostic, and predictive tools.

GBM is commonly stratified into four subtypes: classical, mesenchymal, proneural, and neural. Since each subtype displays different genomic features that affect the resistance mechanisms associated with or modulated by the PC, it has been proposed that PCs could be used as a diagnostic tool to discriminate the different subtypes or to predict the response to targeted therapies [132]. PCs could also serve as a diagnostic tool in medulloblastoma, where the presence or absence of cilia is associated with specific variants. PCs were found in medulloblastomas with activation of HH or Wnt signaling, but not in most medulloblastomas in other distinct molecular subgroups [68].

In glioma, a recent study identified a cilium-associated signature as a prognostic predictive biomarker; in fact, the high expression of the cilium-associated signature has been correlated with poor survival [131]. The authors also introduced a ciliary expression risk score based on the expression of 12 cilium-associated genes as independent prognostic biomarkers [131].

### 4.2. Skin Cancers

The presence of PCs on the cell surface of lesional melanocytes within conventional melanocytic nevi was first reported in 2011 by Kim et al. [135]. An independent group validated the original findings by evaluating PCs by immunofluorescence microscopy on 87 cases of melanocytic nevi and melanomas [181]. Both studies proposed the use of PC expression as a diagnostic tool to identify the early stages of melanomas [135,181].

In melanoma, PCs’ function has been linked to a negative regulation of Wnt pathway oncogenic activity [182]. Importantly, the Wnt pathway can also mediate cancer immune evasion and resistance to immunotherapies [182]. Altogether, these findings suggested the potential use of PC staining as a diagnostic tool for dermatopathologists in ambiguous melanocytic neoplasms or as a predictive marker for the response to immunotherapies.

### 4.3. Gastrointestinal Cancers

The prognostic significance of the frequency of PCs in small bowel and colorectal adenocarcinoma has been assessed by Dvorak and colleagues [140], supporting a potential use for PC as a biomarker in these types of cancer.

Epigenetic alterations of ciliary genes may also be important for diagnostic and prognostic purposes. An altered DNA methylation pattern of ciliary genes has been reported in CCAs [183]. The DNA methylation gene biomarker profile has also been proposed as a highly sensitive and specific diagnostic tool in CCA. Of note, the expression profile of histone deacetylases (HDACs), known as epigenetic regulators of gene expression including the ciliary-associated protein HDAC6, has been investigated as a prognostic biomarker in CCA [57,113,127,183]. The prognostic value of HDAC6 overexpression, which causes PC shortening and promotes cell growth, has been reported in CCAs [57,113,127,183].

The clinical implications of ciliary genes in PDAC have been recently reviewed [177]. In particular, lower expression of ciliary-associated genes *AURKA*, *PLK1*, and *NEK2* and higher expression of ciliary-related HDAC6 and INPP5E were associated with favorable prognosis. The expression of *AURKA*, *PLK1*, *NEK2*, *KIF2A*, *NEDD9*, and calmodulin (*CALM*) 1, 2, and 3 is significantly higher in tumors compared with normal tissues. In particular, the expression profile of *AURKA* has been associated with a negative prognostic index, as *AURKA* is overexpressed in tumors of higher histological grade. On the other hand, the overexpression of the cilioprotein INPP5E has been associated with a favorable prognosis [177].

### 4.4. Genito-Urinary and Endocrine Cancers

Several studies have demonstrated a decreased number of PCs in breast cancer cells and patients [157,158,184]. Menzl et al. reported a significant decrease in PC number by analyzing a large cohort of pre-invasive and invasive breast cancers of low- and high-grade tumors, including Luminal A, Luminal B, Her2+, and Triple-Negative tumors, in all stages and subtypes [157].

In addition, they observed that the PC number is significatively decreased in pre-invasive breast carcinoma in situ (CIS), suggesting that the presence of PCs could be considered a biomarker of this subtype. Of note, the diagnosis and prognosis of CIS represents a clinical challenge since CIS patients have an increased risk of developing invasive breast cancer, but not all patients progress to the invasive stage [157]. Thus, the identification of new prognostic and predictive biomarkers of CIS may potentially address the above-described clinical needs.

Furthermore, the group of Légaré et al. investigated the clinical significance of the use of SPEN as a predictive biomarker of metastatic events [159]. The authors found that the SPEN expression profile is correlated with metastatic events in breast cancer in independent cohorts of ERα-negative tumors [159]. Moreover, the SPEN expression profile is associated with the very low abundance of PCs in luminal and ERα-positive breast cancer cells in vitro [159].

The study of Hassounah and colleagues first characterized the number of PCs in human tissues from different stages of prostate cancer [160]. They examined the correlation between PC expression and Wnt signaling and demonstrated that PC frequency is decreased in all stages of prostate cancer, from early preinvasive lesions to invasive stages. They also investigated the association between PC number and clinical parameters and most notably found that a decreased cilium frequency correlated with increased tumor size, suggesting the potential use as a diagnostic tool in prostate cancer [160].

Interesting, as previously mentioned, *FGFR2-OFD1* gene fusions were reported in TC [164,165]. Of note, *FGFR2* fusions’ detection represents reliable biomarkers to predict the response to FGFR tyrosine kinase inhibitors, recently introduced in the treatment of several cancers [165,185].

Finally, two studies analyzed cilia expression in tissue sections from renal cancer patients and observed a severe reduction of cilia number in RCC subtypes, thus supporting evidence that ciliary dysfunction may be considered a diagnostic biomarker of most renal tumors [161,186].

### 4.5. Implication of PC in Cancer Therapeutics

The etiopathology of a plethora of diseases has been associated with alteration in PC structure and function. Methods to alter cilium number and length have been investigated as potential therapeutic strategies. The term “ciliotherapy” refers to a targeted therapy focused on PCs, and it has been recently introduced as a treatment strategy to target cilia in several genetic diseases known as ciliopathies [179,187,188,189,190]. Intensive pre-clinical studies have revealed new treatment approaches targeting PCs in ciliopathies, providing promising results [179,187,188,189,190].

For example, genome-wide methylation profiling of PKD has identified epigenetically regulated genes that are associated with renal cyst development, suggesting epigenetic therapy as a potential treatment. Due to the molecular similarities between PKD and RCC, these approaches targeting cystogenesis have been proposed as promising therapeutic strategies in cancer [191].

Moreover, several drugs including LiCl and fenoldopam have been investigated to specifically target PCs in patients with PKD [187,188]. Other studies demonstrated the efficacy of pharmacological inhibition of HDAC6 in renal cystic disease and obesity associated with BBS syndrome [189]. The impact of pharmacological inhibition of CDK5 and GLI2 was also evaluated at the pre-clinical level [190].

Accumulating evidence has highlighted the role of PCs in cancer, leading scientists to investigate ciliotherapies as a possible strategy to target cilia in cancer. Data from solid tumors strongly suggested that the PC functions as a tumor suppressor. Therefore, the mechanisms that cancer cells develop to inhibit ciliogenesis give them a selective proliferative advantage. Thus, therapeutic approaches directed towards the restoration of PC expression have been investigated also in cancer.

HDAC6 has been identified as a major driver of ciliary disassembly, so several authors suggested that treatment with HDAC6 inhibitors may lead to increased length and number of PCs, concomitant with a suppression of tumor growth. HDAC6 causes a shortening of PCs through different mechanisms [57]. It is a cytoplasmatic enzyme that mediates deacetylation of α-tubulin and cortactin, two important components of cilia, contributing to microtubule destabilization and ciliary disassembly. HDAC6 inhibition with pharmacological or genetic approaches has been used to induce PC restoration and reverse the malignant phenotype in cholangiocarcinoma (CCA) [57]. Additionally, HDAC6 inhibition also suppresses proliferation and invasion in chondrosarcoma tumor cells and restores the expression of PCs [57]. The use of pan-HDAC inhibitors is controversial considering the potential adverse effects and broad epigenetic changes. Several non-selective HDAC inhibitors have already been approved by the U.S. Food and Drug Administration (FDA) for cancer treatment including vorinostat, romidepsin, and panobinostat, but they show toxicity to normal tissues during cancer therapy. In contrast, specific HDAC6 inhibition is promising due to its wide tolerability. Efforts have been made to increase the specificity of the pharmacological inhibitors of HDAC6, developing compounds based on the structure of its enzymatic active site. The therapeutic use of tubacin, tubastatin A, CAY10603, and ACY1215, previously shown to be efficient also in several ciliopathies including PKD, was shown to be effective in cancer [167]. Furthermore, the study of specific inhibitors of other HDACs has been the object of several investigations. In an orthotopic rat CCA model, the inhibitor of sirtinol (SIRT1) reduced tumor size and tumorigenic protein expression. In vitro and in vivo experiments using SIRT1 inhibitors showed a reduction in tumor growth associated with enhanced ciliary expression, suggesting that the reestablishment of PCs in CCA cells by means of SIRT1 inhibitors may be a potential therapeutic approach [128]. Using in vitro and in vivo models of CCA, chalcones have been described to activate the tumor suppressor activity of LKB1, a tumor suppressor gene involved in the modulation of ciliogenesis [176]. Kobayashi and colleagues demonstrated that inhibition of KRAS and HDAC2 may restore PCs in PDAC cells, suggesting that HDAC2 inhibitors may be proposed as a new ciliotherapy [155]. In line with these results, Ischenko et al. reported that the combination of HDACs and KRAS inhibitors provides an effective strategy for the treatment of PDAC [192].

Khan et al. screened 1600 drugs for their efficacy in restoring PCs and concomitantly suppressing tumor growth [193]. They identified 110 compounds able to increase ciliogenesis and decrease cell proliferation, including clofibrate, gefitinib, sirolimus, imexon, and dexamethasone [193]. Several authors suggested that therapeutic strategies targeting PCs to inhibit oncogenic pathways, including HH, Wnt, and the autophagy–lysosome pathway, could represent promising approaches [194,195,196]. In PDAC, the suppression of ciliogenesis causes upregulation of metabolic-related mevalonate pathway (MVA) enzymes through Wnt-β catenin signaling and induces neoplastic transformation in both mouse models and human samples of PDAC. In line with these findings, the authors also proposed the use of statin as an inhibitor of MVA as a possible treatment of PDAC [156].

The HH pathway promotes cancer growth, so novel drugs that antagonize HH signaling components, such as SMO, could prove of therapeutic value. A high-throughput screen for inhibitors of SMO ciliary localization and ciliogenesis led to the identification of two ciliogenesis antagonists that disrupt ciliogenesis, inhibit the activation of the HH pathway, and abrogate the proliferation of basal-cell-carcinoma-like cancer cells (BCC) [194]. Several clinical trials are evaluating the potential use of SMO inhibitors for a variety of cancer treatments including Cyclopamine, Saridegib, Vismodegib, Cur61414, XL-139, and Sonidegib [197,198,199,200,201]. Vismodegib showed an acceptable safety profile and encouraging anti-tumor activity in advanced BCC and MB [201]. Recent studies have also shown that drugs, including budesonide, which acts by interfering with SMO ciliary trafficking, have been proposed as a novel potential therapeutic approach to inhibit oncogenic HH pathway activation [202].

In glioblastoma, several therapeutic strategies have been proposed to target factors belonging to HH signaling, including PTCH-SMO [132]. O6-methylguanine-DNA methyltransferase (MGMT), a DNA repair enzyme and downstream effector of the HH cascade, has been reported to significantly contribute to the development of drug resistance in both glioma and glioblastoma [172].

In most cell types of the human body, signaling by PCs involves G-protein-coupled receptors (GPCRs), which transmit specific signals to the cell through G proteins to regulate cellular and physiological events. Historically, GPCRs have been demonstrated to be potent targets of a large group of drugs [195]. High-throughput screening campaigns have successfully identified potent GPCR drug candidates, suggesting the possibility that selective manipulation of converging GPCR pathways specifically in the ciliary compartment could be feasible [196]. These studies promise to offer key insights into the role of PCs as critical pharmacological targets in the treatment of cancer.

A recent report identified lysophosphatidic acid receptor (1LPAR1) as a therapeutic target in GBM [173]. The same authors demonstrated that LPAR1 may drive GBM proliferation in a cilia-dependent manner and proposed a therapeutic targeting strategy against LPAR1 [173].

In several cancers, including GBM, the overexpression of cell-cycle-related kinase (CCRK) and its substrate intestinal cell kinase (ICK) inhibits ciliogenesis and promotes tumor growth [174]. Further analysis revealed that CCRK is involved in the modulation of signaling networks comprising AR, Wnt, AKT, EZH2, NF-κB, and HH, and CCRK inhibitors have been found to suppress tumor proliferation [174,175].

EZH2 has been shown to promote tumorigenesis by suppressing PC genes, and numerous EZH2 inhibitors have entered clinical trials [203]. In melanoma, the EZH2 gain in benign melanocytic lesions has been shown to promote the loss of PCs, to enhance pro-tumorigenic Wnt/β-catenin signaling, and to drive metastatic processes. Specific inhibitors of EZH2 activity induce ciliogenesis and cilia-dependent tumor growth suppression and are, thus, considered as a strategy for treating melanoma [136].

Cyclin-dependent kinases (CDKs) have emerged as key primary ciliogenesis regulators [179]. In earlier studies, the efficacy of CDK5 inhibitors was assessed in PKD mouse models [179]. Notably, CDK5 participates in numerous tumorigenic processes. Pre-clinical and early clinical studies have confirmed that CDK5 inhibitors are potent anticancer agents [178].

A proposed approach to target the PC is by the inhibition of ciliogenesis-associated kinase 1 (CILK1), a negative regulator of cilia length [180]. Alvocidib, a semi-synthetic flavone related to a natural product extracted from Indian plants, has been shown to be a potent inhibitor of CILK1 in cells, suggesting a potential use for targeting PCs in cancer [180].

## 5. Conclusions

Cilia are organelles present in the majority of mammalian cells with a high level of conservation across a variety of species. These two main features suggest an important role in development and normal physiological functions. Recent studies have taken this organelle from obscurity to the forefront of cutting-edge research, showing its importance in human diseases including ciliopathies and cancer. Several studies have demonstrated that the primary cilium (PC) takes part in two principal molecular processes: (a) a plethora of external stimuli converges on PCs, which works as a signaling hub, participating in signal transduction inside the cell; (b) ciliogenesis and mitosis are two processes mutually interconnected; this overlap makes PCs an important regulator of the cell cycle and cell survival.

In this review, we discussed the involvement of PCs in cancer at a molecular level, highlighting the implications of altered cilia-related factors at the clinical level, as well as the more-recent therapeutic strategies designed to target the PC or PC-related molecular components. Further studies are required to understand the complexity of the ciliome and identify cilia-related therapeutic targets to be introduced in clinical trials.

Understanding and manipulating PCs not only in cancer cells, but also in cells of the TME could identify novel cancer pathogenetic factors and related anticancer therapies. Among future topics of interest are: (a) ectosome-mediated signaling, which passes through and is driven and regulated by the PC; (b) the modulation of immune system functions operated by the PC; (c) the molecular links between PCs and cell metabolism.

The use of ciliotherapy in cancer is a promising strategy that is still under preclinical study, and the relevance of these therapeutic approaches is under discussion. Instead, the use of PC-related factors as diagnostic, predictive, or prognostic biomarkers are at a more advanced stage in cancer and cancer therapeutics.

In the emerging era of individualized medicine, further investigation is required to fully characterize the ciliome in cancer and transfer this information to clinical studies to evaluate the potential use of PCs as candidate targets or biomarkers in cancer.

## Figures and Tables

**Figure 1 genes-14-01428-f001:**
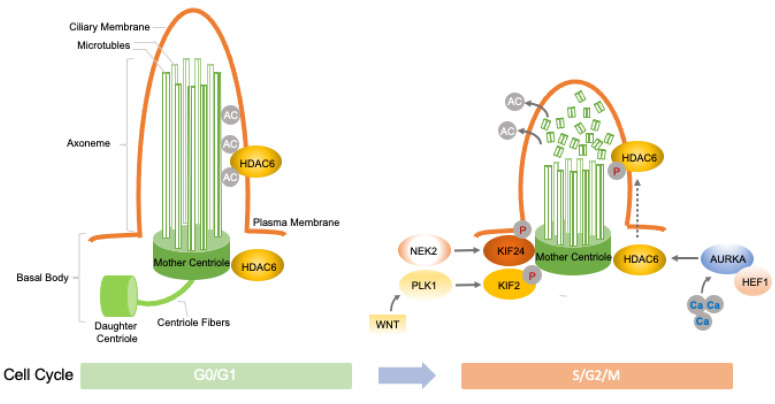
Regulation of ciliogenesis by inputs from the cell cycle. Cilium length is indicated by axonemal microtubules (grey rods) bound by the ciliary membrane (orange line) throughout the cell cycle in proliferating cells (S, G2, M, and G1 phases) and during exit into and entry from cell quiescence (G0 phase). Acetyl groups (Acs) shown as grey circles. The various proteins that are discussed in the main text are highlighted in different colors. “P” in a grey circle indicates protein phosphorylation.

**Figure 2 genes-14-01428-f002:**
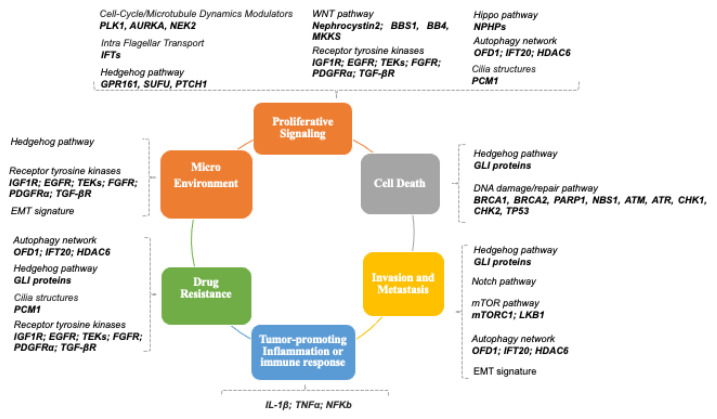
Linking ciliome to cancer hallmarks. Distribution of ciliary-related genes and respective signaling pathways among the categories of cancer hallmarks.

**Table 1 genes-14-01428-t001:** Clinical implications of primary cilia in cancer.

Gene/Gene Signatures	Molecular Pathway	Type of Cancer	Clinical Implication	References
*PKD1; PKD2*	Wnt; RTK; Mechanotransduction	Colorectal, Melanoma	Therapeutic Targeting	[118]
*PTCH1; CTNNB1*	HH; Wnt	Medulloblastoma	Diagnostic biomarkers	[68]
*SHH; PTCH1; SMO; GLI*	HH	Glioblastoma	Therapeutic Targeting	[132]
*MGMT*	HH	Glioblastoma/Glioma	Drug Resistance	[132,172]
*LPAR1*	GPCR	Glioblastoma	Therapeutic Targeting	[173]
*CCRK*	AR, Wnt, AKT, EZH2, and NF-κB, HH	Glioblastoma	Therapeutic Targeting	[174,175]
*LRGUK, NSUN7, LRRC27, SPAG17, EFHB, IFT27, DZIP1L, FOLR1, RGS22, TEX9, GALNT3, and GLB1L*	Cilium-Associated Genes	Glioma	Prognostic Biomarkers	[131]
*EZH2*	Wnt/b-Catenin	Melanoma	Therapeutic Targeting; Diagnostic Biomarkers	[136]
*PTCH, SMO; GLI1*	HH	Colorectal Cancer	Diagnostic and Predictive Biomarkers	[141]
*IFT88*	HH; MAPK	Cholangiocarcinoma; Thyroid Cancers	Diagnostic Biomarker	[176]
*HDAC6*	HH; MAPK	Cholangiocarcinoma; Chondrosarcoma	Therapeutic Targeting; Diagnostic, Predictive Biomarkers	[57,113]
*SIRT1*	HH; AKT; IL6	Cholangiocarcinoma	Therapeutic Targeting	[128]
*HDAC2*	KRAS	Pancreatic Ductal Adenocarcinoma	Therapeutic Targeting	[155]
*AURKA; INPP5E*	Cilium-Associated Genes	Pancreatic Ductal Adenocarcinoma	Prognostic Biomarkers	[177]
*SPEN*	ERα	Breast Cancer	Prognostic, predictive Biomarkers	[159]
*CDK5*	Cell-Cycle-Related	Several Cancers	Prognostic, predictive Biomarkers; Therapeutic targeting	[178,179]
*CILK1*	HH	Several Cancers	Therapeutic targeting	[178,180]

## Data Availability

Not applicable.

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
