# Peer review of "Cilia and Cancer: From Molecular Genetics to Therapeutic Strategies"

_genes, 2023, doi:10.3390/genes14071428_

Round 1

Reviewer 1 Report

The review by Carotenuto et al is a timely treatise on the role of the primary cilium in oncogenesis. The authors describe the roles of primary cilia in promoting or inhibiting oncogenesis based on known or predicted functions of ilia and cilia-associated proteins in several key processes and related clinical implications.  The manuscript describes role of primary cilia in oncogenic programs with relation to cell cycle, proliferative signaling and cancer hallmarks. Different cilia-regulated pathways are described as are different cancer subtypes. Finally, implication of cilia in cancer therapeutics is described. Overall, the review covers a lot of ground ranging from molecular genetics to therapeutic strategies.

A few comments are mentioned below.

  • While GPR161 overexpression has been described in triple negative breast cancer as the authors mention (line 178), GPR161 also functions as a tumor suppressor in medulloblastoma including Gorlins syndrome (PMID: 29386106 and 31609649).
  • SUFU, another negative regulator of Hedgehog signaling that localizes to cilia tips, also has broad cancer relevance including Gorlins syndrome (eg PMID: 25403219).
  • Role of cilia in PKD has been reviewed elsewhere and can be discussed with respect to cilia’s role in proliferation, and therapeutics (PMID: 32475690, 35832738).
  • In therapeutics section, SMO inhibitors such as Vismodegib can be described.

None.

Author Response

REV#1

=

We thank the reviewer for the advice and comments. Below are the point by point rebuttal:

  • We revised the paragraph regarding the HH pathways, introducing a more comprehensive discussion regarding HH regulators, including GPR161, SUFU (line 214-228) and relative references.
  • We introduced a new section (2.2.8) discussing the Role of cilia in PKD.
  • We also reported some findings regarding the therapeutic use of Vismodegib in cancer (line 753-765).

Reviewer 2 Report

This is a good review, which is very thorough and goes into a lot of depth regarding cilia and links to cancer, highlighting up to-date research in an exciting and emerging field. 

More references need to be added, specifically in the introduction and section 2.1. 

Sensory cilia. – The term sensory cilia used in the abstract and the in the introduction is too vague. In humans, there are primary cilia, motile cilia, and specialised cilia such as the photoreceptor connecting cilium and olfactory cilia which also have ‘sensory’ properties. Additionally, there is now some literature that could suggest that primary cilia have the potential to signal themselves, emitting extracellular vesicles. It seems more accurate if PC are referred to as signal transducer or a signal transduction hub etc. 

Introduction needs some work. It is greatly lacking references (only 3 references throughout). The structure of the introduction needs a better flow, it bounces a lot between structure of cilia types, then a little bit about disease, then back to structure. It would read better if the structure and function of each cilia type is mentioned first, before than moving onto clinical implications in ciliary diseases, and then introduce the link with cancer. 

1Specialised cilia are not mentioned in your introduction. It would benefit for this to be briefly mentioned, such as nodal cilia and photoreceptor connecting cilium, as these have major implications for the ciliopathy disease spectrum.  

1   Figure legends should be expanded to further describe each figure.

1   In section 2.2, how each pathway is linked to cancer is sometimes not mentioned (i.e 2.2.3 Wnt, 2.2.6 DNA damage repair). Could this be rectified?

      It would be good to add to Table 1 (or create a new table) to summarise section 3, describing each type of cancer, whether the PC is lost or remains, what PC signalling pathway is linked to the cancer, and whether it is upregulated or downregulated.

1    In terms of therapeutic treatments, it would be good to review the current pitfalls and potential side effects of targeting core cilia pathways, such as HH, and what is the future for targeting the PC in cancer treatment, and why it offers more/different opportunities than current cancer treatments.

1   It would be good to clarify the genetics of ciliopathy disease (rare, largely autosomal recessive, some autosomal dominant). Could the authors discuss whether mutations in ciliary genes are thought to be the initiators/drivers of some cancers (and could this be exploited) and do ciliopathy patients have cancers? Or, is the cancer further enabled by cilia function, and therefore the PC is a target to ameliorate cancer.  

Please see specific comments attached. 

Quality of English is fine, there were only a few grammatical errors. 

Author Response

We thank the reviewer for the advice and comments. Below are the point by point rebuttal:

  • We revised the introduction as suggested by the reviewer, adding mentions about nodal cilia and photoreceptor and major implications for the ciliopathy disease spectrum
  • We added to the text a mention regarding how pathways are linked to cancer
  • We thank the reviewer for the comment, but a similar table has been reported by other authors (https://doi.org/10.1186/s13630-016-0035-3; 2174/1389450116666150223162737) and it would seem repetitive
  • We added more data and discussed more about the targeting core cilia pathways, such as HH, and what is the future for targeting the PC
  • The concept that genetics of ciliopathy diseases and mutations in ciliary genes are thought to be the initiators/drivers of some cancers have been now extensively cited and discussed in the text
  • All the specific comments of rev#2 have been addressed.